# First report of *bla*$_{OXA-181}$-carrying IncX3 plasmids in multidrug-resistant *Enterobacter hormaechei* and *Serratia nevei* recovered from canine and feline opportunistic infections

Chavin Leelapsawas,[1] Parinya Sroithongkham,[1] Sunchai Payungporn,[2] Pattaraporn Nimsamer,[2] Jitrapa Yindee,[1] Alexandra Collaud,[3] Vincent Perreten,[3] Pattrarat Chanchaithong[1,4]

**ABSTRACT**   Whole-genome sequence analysis of six *Enterobacter hormaechei* and two *Serratia nevei* strains, using a hybrid assembly of Illumina and Oxford Nanopore Technologies sequencing, revealed the presence of the epidemic *bla*$_{OXA-181}$-carrying IncX3 plasmids co-harboring *qnrS1* and Δ*ere*(A) genes, as well as multiple multidrug resistance (MDR) plasmids disseminating in all strains, originated from dogs and cats in Thailand. The subspecies and sequence types (ST) of the *E. hormaechei* strains recovered from canine and feline opportunistic infections included *E. hormaechei* subsp. *xiangfangensis* ST171 (*n* = 3), ST121 (*n* = 1), and ST182 (*n* = 1), as well as *E. hormaechei* subsp. *steigerwaltii* ST65 (*n* = 1). Five of the six *E. hormaechei* strains harbored an identical 51,479-bp *bla*$_{OXA-181}$-carrying IncX3 plasmid. However, the *bla*$_{OXA-181}$ plasmid (pCUVET22-969.1) of the *E. hormaechei* strain CUVET22-969 presented a variation due to the insertion of IS*Kpn74* and IS*Sbo1* into the *virB* region. Additionally, the *bla*$_{OXA-181}$ plasmids of *S. nevei* strains were nearly identical to the others at the nucleotide level, with IS*Ecl1* inserted upstream of the *qnrS1* gene. The *E. hormaechei* and *S. nevei* lineages from canine and feline origins might acquire the epidemic *bla*$_{OXA-181}$-carrying IncX3 and MDR plasmids, which are shared among Enterobacterales, contributing to the development of resistance. These findings suggest the spillover of significant OXA-181-encoding plasmids to these bacteria, causing severe opportunistic infections in dogs and cats in Thailand. Surveillance and effective hygienic practice, especially in hospitalized animals and veterinary hospitals, should be urgently implemented to prevent the spread of these plasmids in healthcare settings and communities.

**IMPORTANCE**   *bla*$_{OXA-181}$ is a significant carbapenemase-encoding gene, usually associated with an epidemic IncX3 plasmid found in Enterobacterales worldwide. In this article, we revealed six carbapenemase-producing (CP) *Enterobacter hormaechei* and two CP *Serratia nevei* strains harboring *bla*$_{OXA-181}$-carrying IncX3 and multidrug resistance plasmids recovered from dogs and cats in Thailand. The carriage of these plasmids can promote extensively drug-resistant properties, limiting antimicrobial treatment options in veterinary medicine. Since *E. hormaechei* and *S. nevei* harboring *bla*$_{OXA-181}$-carrying IncX3 plasmids have not been previously reported in dogs and cats, our findings provide the first evidence of dissemination of the epidemic plasmids in these bacterial species isolated from animal origins. Pets in communities can serve as reservoirs of significant antimicrobial resistance determinants. This situation places a burden on antimicrobial treatment in small animal practice and poses a public health threat.

**KEYWORDS**   *bla*$_{OXA-181}$, carbapenemase, cats, dogs, Enterobacterales, MDR, WGS

Address correspondence to Pattrarat Chanchaithong, Pattrarat.C@chula.ac.th.

The authors declare no conflict of interest.

See the funding table on p. 5.

Carbapenem-resistant Enterobacterales are Gram-negative bacteria of major public health concern, causing a wide range of opportunistic infections in hospitals. Alongside *Escherichia coli* and *Klebsiella pneumoniae*, the *Enterobacter cloacae* complex (ECC) and *Serratia* species are increasingly recognized as causing various opportunistic diseases in hospitalized patients (1, 2) and are occasionally associated with opportunistic infections in dogs and cats (3, 4). In addition to their intrinsic resistance, they can become multidrug-resistant (MDR) by acquiring antimicrobial resistance genes (ARGs) on mobile genetic elements, limiting therapeutic options (5–7). Carbapenems have been off-label prescribed in the treatment of small animals in veterinary medicine (8). In recent years, several publications have reported the dissemination of carbapenemase-encoding genes associated with specific plasmid families in ECC and *Serratia* spp. in humans, companion animals, and veterinary settings, but so far, none of the veterinary strains were found to carry $bla_{OXA-181}$ (7, 9, 10).

Six carbapenemase-producing (CP) ECC and two CP *Serratia marcescens* strains, identified using matrix-assisted laser desorption ionization time-of-flight mass spectrometry (Bruker Daltonics GmbH, Germany), were isolated from canine and feline diagnostic samples sent to the Veterinary Diagnostic Laboratory, Faculty of Veterinary Science, Chulalongkorn University, Bangkok, Thailand, between 2018 and 2022. Whole-genome-based identification using BLAST-based average nucleotide identity through the online available service JSpeciesWS (https://jspecies.ribohost.com/jspeciesws/) reclassified the ECC as *Enterobacter hormaechei* subsp. *xiangfangensis* ($n = 5$) and *E. hormaechei* subsp. *steigerwaltii* ($n = 1$), and the *S. marcescens* as *Serratia nevei* (Table S1). The two CP *S. nevei* (CP-*Sn*) and three CP *E. hormaechei* (CP-*Eh*) strains were recovered from dogs, while the other three CP-*Eh* strains were isolated from cats. Carbapenem resistance and the presence of $bla_{OXA-48-like}$ gene were first observed with imipenem minimum inhibitory concentration $\geq$16 µg/mL in diagnostic results using Vitek 2 automated system (bioMérieux, France) and by multiplex PCRs, respectively (11, 12).

Antimicrobial susceptibility testing was further conducted by broth microdilution assay using Sensititre ASSECAF/ASSECB plates (Thermo Fisher Scientific, USA). Resistance was interpreted following the interpretive criteria of the Clinical and Laboratory Standards Institute, except for tigecycline, which was referred to the criteria of *E. coli* from the European Committee on Antimicrobial Susceptibility Testing (13–15). The two CP-*Sn* isolates exhibited resistance to all drugs tested. Antimicrobial resistance phenotypes of the six CP-*Eh* strains were variable (Table S2). However, all CP-*Eh* isolates exhibited resistance to ampicillin, meropenem, nalidixic acid, ciprofloxacin, and azithromycin but were still susceptible to colistin and tigecycline.

Whole-genome sequences of all strains were obtained using the hybrid assembly of sequence reads from both Illumina Nextseq 550 and Oxford Nanopore Technologies. Raw reads were qualified using the FastQC program v0.11.9 (https://www.bioinformatics.babraham.ac.uk/projects/fastqc/). Illumina reads and sequence adaptors were trimmed and removed using Trimmomatic v0.39. Assembly of the reads was done using Unicycler v0.4.8 generating contigs of circular complete genomes and plasmids in all strains (Table S3). The quality of the contigs was assessed using QUAST v5.0.2 (https://github.com/ablab/quast) before submission to the NCBI Prokaryotic Genome Annotation Pipeline for gene annotation (Table S3). Whole-genome sequence analysis revealed the presence of $bla_{OXA-181}$ gene and additional ARGs (Table S1) in all strains by using NCBI AMRFinderPlus v3.11.17, Resfinder v4.1, and CARD v3.2.7. Plasmid incompatibility complex (Inc) groups and insertion sequence (IS) elements were identified by PlasmidFinder v2.1 and ISFinder tools, respectively. Three of the CP-*Eh* strains belong to the ST171 high-risk clone (16), while the others were ST65, ST121, and ST182. The core genome of CP-*Eh* was extracted using the Roary pipeline v3.13.0 before performing single-nucleotide polymorphism (SNP) analysis of the core gene alignment to obtain core genome single-nucleotide polymorphism (cgSNP) using SNP-sites v2.5.1. cgSNP-based phylogeny obtained using RaxML v8.2.11 revealed that, within the ST171 clades, CUVET21-1190 and CUVET21-1726 had identical core genes and differed from

CUVET22-969 by only 270 SNPs (99.8% identity). The CP-*Eh* ST65, ST121, and ST182 formed different clades with a difference of >12,644 SNPs. The cgSNP was not detected between two CP-*Sn* strains (Fig. 1). Genes associated with virulence and adaptation detected by using virulence factor database (http://www.mgc.ac.cn/VFs/) were present in the chromosome and plasmid of both *E. hormaechei* and *S. nevei* (Fig. S1). The CP-*Eh* strains exhibited a higher number of virulence-associated genes compared to other *Enterobacter* species or strains in a previous study (17). The abundance of chromosomal virulence-associated genes in these bacteria may play important roles in survival within the environment and animal intestinal tract.

The OXA-181-producing *E. hormaechei* and *S. nevei* strains carried two to eight plasmids. The $bla_{\text{OXA-181}}$-carrying IncX3 plasmid was present in all strains and co-harbored *qnrS1* and Δ*ere*(A) genes (Fig. 2). Five of the six CP-*Eh* strains carried an identical 51,479-bp $bla_{\text{OXA-181}}$-carrying plasmid, which shared 100% DNA identity to an *E. hormaechei* plasmid (pM206-OXA-181; GenBank accession number AP018831) isolated from a Burmese patient (18). The 54,232-bp IncX3 plasmid pCUVET22-969.1 of strain CUVET22-969 had a larger size than the others due to the interruption of *virB1* by IS*Sbo1* and the insertion of IS*Kpn74* between *virB10* and *virB11*. The IS*Kpn74* had two 17-bp inverted repeats, each preceded by a 9-bp direct repeat (DR). The presence of IS originally identified in other Enterobacterales and their absence in the CP-*Eh* ST171 genomes suggest that plasmid pCUVET22-969.1 has been transiting in other bacteria prior acquisition by *E. hormaechei*, also highlighting the promiscuous potential of IncX3 plasmids for carbapenemase dissemination. The 52,830-bp IncX3 plasmids pCUVET18-1371.4 and pCUVET18-1784.4 of CP-*Sn* strains had a few SNPs compared to pM206-OXA-181 and had an additional insertion of IS*Ecl1* upstream of *qnrS1*, flanked by two 14-bp DRs (Fig. 2). ISs can function as molecular hotspots, facilitating the acquisition of additional genes and promoting recombination for the evolution of plasmids (19). The resistance genes were situated in a 14-kbp IS*26* pseudo-composite transposon, located downstream of *umuD* (20, 21). The $bla_{\text{OXA-181}}$ was preceded by ΔIS*Ecp1* and followed

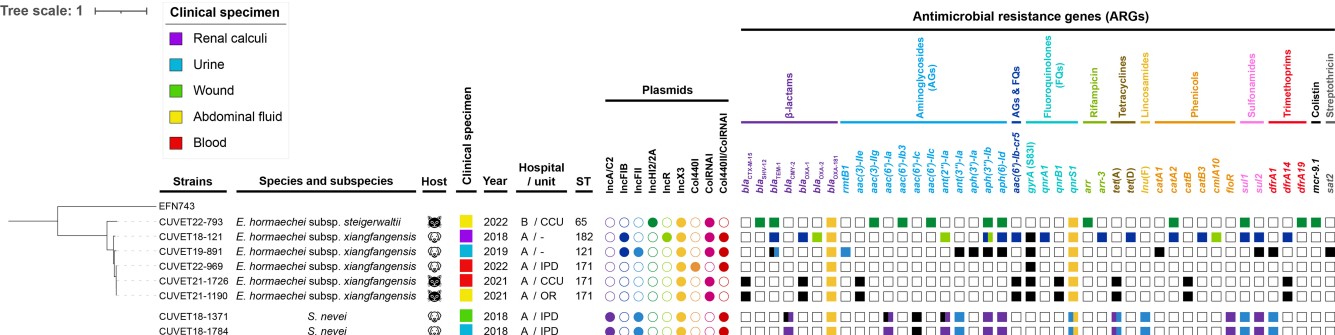

**FIG 1** Phylogenetic tree constructed from cgSNP of the six OXA-181-producing *Enterobacter hormaechei* strains and data from two strains of OXA-181-producing *Serratia nevei* isolated from dog and cat patients entering two small animal hospitals (hospital A and B) in Bangkok, Thailand, along with metadata. The tree was rooted with the OXA-181-producing *E. cloacae* strain EFN743 (GenBank accession number CP092635–CP092638). The empty circles and squares denote the absence of plasmids or ARGs. The presence of ARGs conferring resistance to each antimicrobial drug class was indicated corresponding to the colored circles of plasmids found in each strain, except for black squares that represent ARGs found on the chromosome. The two colors in the same square indicated that the ARG was found on two plasmids or on a plasmid and the chromosome. Unit in hospitals: –, not indicated; CCU, critical care unit; IPD, in-patient department; OR, operating room. ARGs: *bla*, beta-lactamases for beta-lactam resistance; *rmtB1*, 16S ribosomal RNA methyltransferase for aminoglycoside resistance; *aac(3)-IIe*, *aac(3)-IIg*, *aac(6')-Ia*, *aac(6')-Ib3*, *aac(6')-Ic*, *aac(6')-IIc*, aminoglycoside acetyltransferase; *ant(2")-Ia*, *ant(3")-Ia*, aminoglycoside nucleotidyltransferase; *aph(3')-Ia*, *aph(3")-Ib*, *aph(6)-Id*, aminoglycoside phosphotransferase; *aac(6')-Ib-cr5*, a fluoroquinolone-acetylating aminoglycoside acetyltransferase; *gyrA* (S83I), a point mutation at position 83 in the quinolone resistance-determining region of *gyrA* where serine (S) was substituted by isoleucine (I) for fluoroquinolone resistance; *qnrA1*, *qnrB1*, *qnrS1*, DNA gyrase protection genes for quinolone resistance; *arr*, *arr-3*, ADP-ribosyltransferase for rifampicin resistance; *tet*(A), *tet*(D), tetracycline efflux genes for tetracycline resistance; *lnu*(F), lincosamide nucleotidyltransferase for lincosamide resistance; *catA1*, *catA2*, *catB*, *catB3*, *cmlA10*, chloramphenicol efflux genes for chloramphenicol resistance; *floR*, florfenicol/chloramphenicol efflux gene for florfenicol and chloramphenicol resistance; *sul1*, *sul2*, dihydropteroate synthase for sulfonamide resistance; *dfrA1*, *dfrA14*, *dfrA19*, dihydrofolate reductase for trimethoprim resistance; *mcr-9.1*, phosphoethanolamine transferase for colistin resistance; *sat2*, streptothricin acetyltransferase for streptothricin resistance.

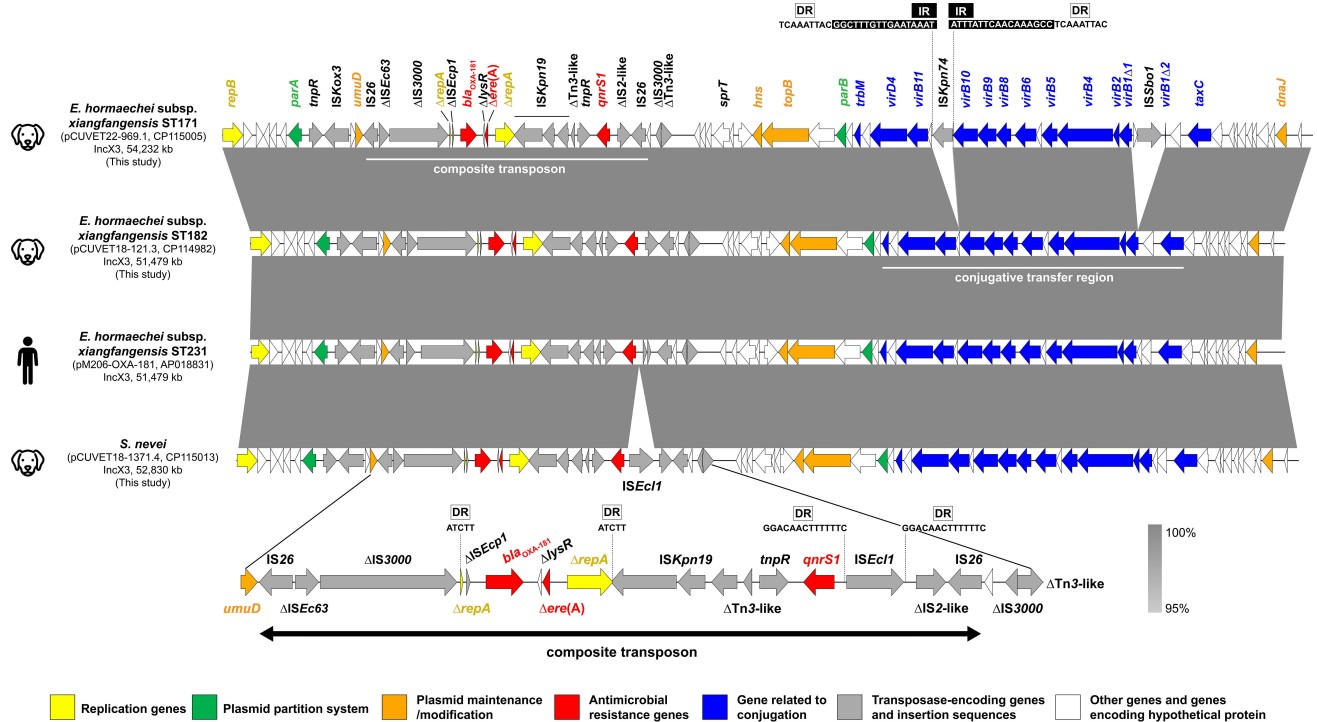

**FIG 2** Sequence comparison of the $bla_{OXA-181}$-carrying IncX3 plasmids in *Enterobacter hormaechei* and *Serratia nevei* isolated from dogs in Thailand with pM206-OXA181 (GenBank accession number AP018831) harbored by *E. hormaechei* ST231 isolated from a patient in Myanmar. The same sequence structure of IncX3 plasmid in *E. hormaechei* ST182 (pCUVET18-121.3) was found in pCUVET19-891.3, pCUVET21-1190.1, pCUVET21-1726.1, and pCUVET22-793.2, all exhibiting 100% nucleotide identity. The IncX3 plasmids (pCUVET18-1371.4 and pCUVET18-1784.4) in two *S. nevei* strains were identical and had an additional insertion of IS*Ecl1* upstream of *qnrS1*, flanked by two 14-bp DRs. The genetic environment of $bla_{OXA-181}$ flanked by two copies of IS*26* composite transposon is demonstrated at the bottom. The 5-bp AT-rich duplicate sequences (ATCTT) of the extremities of Δ*repA* suggested insertion and subsequent recombination between two copies of IS*Ecp1* (20, 22). The arrows indicate the orientation and length of genes. The color of each arrow represents the gene type or function as given in the legend below. The dark gray shading indicates 100% sequence similarity. The symbol Δ indicates truncation.

by Δ*lysR* and Δ*ere*(A), which were flanked by fragments of *repA*. A 5-bp AT-rich sequence repeat (ATCTT) was present at the extremities of Δ*repA*, suggesting the insertion and subsequent recombination between two copies of IS*Ecp1* of the ColKP3 replicon into this region (20, 22). The *qnrS1* was separately located downstream of Δ*repA* at the right extremity of the transposon. Additionally, these plasmids were mobilizable, as they had conjugative regions consisting of a relaxase gene (*taxC*), a gene encoding type IV coupling protein (*virD4*), and *virB* gene cluster of type IV secretion system (Fig. 2) (18). The broad host range ability of IncX3 plasmids has contributed to the widespread dissemination of carbapenemase-encoding genes, including $bla_{OXA-181}$ and $bla_{NDM-5}$, in Enterobacterales (23). Identical or highly similar IncX3 plasmids have also been identified in *E. coli*, *K. pneumoniae*, and *S. marcescens* (20, 21, 24, 25). The detection of this resistance plasmid in *Enterobacter* and *Serratia* is relatively rare, as these genera causing opportunistic diseases are less commonly found when compared to *E. coli* and *K. pneumoniae* (2, 17).

The canine and feline *E. hormaechei* and *S. nevei* strains were MDR and carried multiple ARGs on both the chromosome and plasmids. Two additional MDR plasmids were detected in CP-*Eh* ST182 strain, including IncFIB and IncR plasmids, which contained 13 and four ARGs, respectively (Table S1). The CP-*Eh* ST65 strain carried a 283,562-bp IncHI2/2A plasmid (pCUVET22-793.1) containing *mcr-9.1*, but the strain was susceptible to colistin. This plasmid was similar to pC45_001 (GenBank accession number CP042552) and pCM18-216 (GenBank accession number CP050312) of *E. hormaechei* ST133 and ST110, respectively, which were isolated from a patient and

hospital environment in Australia (Fig. S2). This *mcr-9.1*-carrying IncHI2/2A plasmid has also been found in other Enterobacterales (7, 26, 27). However, the expression of *mcr-9.1* in this pCUVET22-793.1 plasmid likely did not occur due to the absence of *qseB−qseC* two-component regulatory genes (28). Additionally, two CP-*Sn* strains harbored identical IncA/C2 and IncFII MDR plasmids, each containing eight and six ARGs, respectively (Table S1).

In summary, our study represents the first detection and description of *E. hormaechei* and *S. nevei* carrying the epidemic $bla_{OXA-181}$-bearing IncX3 plasmids originating from animals. This also indicates that animals and animal environment contribute to the dissemination of MDR and carbapenemase-encoding plasmids among different species of opportunistic Enterobacterales. Given their proximity to humans and communities, monitoring the spread of clonal lineages and plasmids, especially in hospitalized animals and veterinary hospitals, should be further implemented.

## ACKNOWLEDGMENTS

This study was funded by the Chulalongkorn University–Veterinary Science Research Fund (RI6/2566) and internal funds from the Institute of Veterinary Bacteriology, University of Bern (REF-660-50). This project was also partially supported by the National Research Council of Thailand Project ID N42A660897. The PhD scholarship for Chavin Leelapsawas was supported by the Second Century Fund (C2F) of Chulalongkorn University.

We would like to express our gratitude to Professor Rungtip Chuanchuen from the Department of Veterinary Public Health, Faculty of Veterinary Science, Chulalongkorn University, for providing the customized Sensititre ASSECAF/ASSECB plates; Dr. Komkiew Pinpimai from Aquatic Resources Research Institute, Chulalongkorn University; and Dr. Michael Brilhante and Javier Eduardo Fernandez from Institute of Veterinary Bacteriology, Vetsuisse Faculty, University of Bern, for providing technical guidance on whole-genome sequencing and analysis.

## AUTHOR AFFILIATIONS

[1]Department of Veterinary Microbiology, Faculty of Veterinary Science, Chulalongkorn University, Bangkok, Thailand
[2]Center of Excellence in Systems Microbiology (CESM), Department of Biochemistry, Faculty of Medicine, Chulalongkorn University, Bangkok, Thailand
[3]Division of Molecular Bacterial Epidemiology and Infectious Diseases, Institute of Veterinary Bacteriology, Vetsuisse Faculty, University of Bern, Bern, Switzerland
[4]Research Unit in Microbial Food Safety and Antimicrobial Resistance, Faculty of Veterinary Science, Chulalongkorn University, Bangkok, Thailand

## AUTHOR ORCIDs

Chavin Leelapsawas  http://orcid.org/0000-0002-7835-2191
Vincent Perreten  http://orcid.org/0000-0001-5722-9445
Pattrarat Chanchaithong  http://orcid.org/0000-0002-8979-2440

## FUNDING

| Funder | Grant(s) | Author(s) |
| --- | --- | --- |
| CU \| Faculty of Veterinary Science, Chulalongkorn University (CU-VET) | RI6/2566 | Pattrarat Chanchaithong |
| UB \| Institute of Veterinary Bacteriology, University of Bern (IVB) | REF-660-50 | Vincent Perreten |
| Chulalongkorn University (CU) | Second Century Fund (C2F) | Chavin Leelapsawas |

| Funder | Grant(s) | Author(s) |
|---|---|---|
| National Research Council of Thailand (NRCT) | N42A660897 | Pattrarat Chanchai-thong |

## AUTHOR CONTRIBUTIONS

Chavin Leelapsawas, Conceptualization, Data curation, Formal analysis, Investigation, Methodology, Software, Validation, Visualization, Writing – original draft | Parinya Sroithongkham, Formal analysis, Investigation, Methodology, Software | Sunchai Payungporn, Conceptualization, Methodology, Resources, Supervision | Pattaraporn Nimsamer, Investigation, Software | Jitrapa Yindee, Data curation, Investigation | Alexandra Collaud, Investigation, Resources, Software | Vincent Perreten, Conceptualization, Funding acquisition, Investigation, Methodology, Resources, Supervision, Validation, Writing – review and editing | Pattrarat Chanchaithong, Conceptualization, Data curation, Funding acquisition, Investigation, Methodology, Project administration, Resources, Supervision, Validation, Visualization, Writing – review and editing

## DATA AVAILABILITY

The genome sequences of six *E. hormaechei* strains and two *S. nevei* strains are deposited in the NCBI database under BioProject accession number PRJNA912905.

## ADDITIONAL FILES

The following material is available online.

### Supplemental Material

**Supplemental material (Spectrum03589-23-S0001.pdf).** Tables S1 to S3; Fig. S1 and S2.

### Open Peer Review

**PEER REVIEW HISTORY (review-history.pdf).** An accounting of the reviewer comments and feedback.

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
