## [Reviewer comments · Microbiology Spectrum]

Microbiology Spectrum

First Report of *bla*_{OXA-181}-Carrying IncX3 Plasmids in Multidrug-Resistant *Enterobacter hormaechei* and *Serratia nevei* Recovered from Canine and Feline Opportunistic Infections

Chavin Leelapsawas, Parinya Sroithongkham, Sunchai Payungporn, Pattaraporn Nimsamer, Jitrapa Yindee, Alexandra Collaud, Vincent Perreten, and Pattrarat Chanchaithong

Corresponding Author(s): Pattrarat Chanchaithong, Chulalongkorn University Faculty of Veterinary Science

Review Timeline:

Submission Date:	October 13, 2023
Editorial Decision:	November 23, 2023
Revision Received:	December 19, 2023
Accepted:	January 14, 2024

Editor: Ruth Hall

Reviewer(s): The reviewers have opted to remain anonymous.

Transaction Report:

DOI: <https://doi.org/10.1128/spectrum.03589-23>

Re: Spectrum03589-23 (First Report of *bla*_{OXA-181}-Carrying IncX3/ColKP3 Plasmids in Multidrug-Resistant *Enterobacter hormaechei* and *Serratia nevei* Recovered from Canine and Feline Opportunistic Infections)

Dear Dr. Pattrarat Chanchaithong:

Thank you for the privilege of reviewing your work. Below you will find my comments, instructions from the Spectrum editorial office, and the reviewer comments.

Please revise in the light of reviewer comments below and in the attached files. In addition please release the sequence reads and provide information about the sequence quality as a Supplementary file.

Revision Guidelines

Sincerely,
Ruth Hall
Editor
Microbiology Spectrum

Reviewer #1 (Comments for the Author):

The manuscript reports interesting data on the dissemination of multi-drug resistance plasmids in enteric bacteria recovered from companion animals in Thailand. Considering the limited information regarding plasmid circulation in animals, this work despite its descriptive nature contributes to a better understanding of AMR epidemiology in these reservoirs. The genomic

analysis is detailed and well presented. I suggest providing some information on sequencing metrics and a better description in the text of the other MDR plasmids identified in these isolates instead of just listing them in Table S1. Find some other minor suggestions in the annotated pdf.

Reviewer #2 (Comments for the Author):

Leelapsawas et al. describe the presence of IncX3-blaOXA-181 in *E. hormaechei* (n=6) and *S. nevei* (n=2) in dogs and cats in Thailand for the first time. Data are generally presented coherently but analysis and discussion would benefit if some modifications were implemented. Please find below some comments for the authors to consider.

1. The continuing reference to IncX3/ColKP3 plasmid in the title and throughout the manuscript should be avoided. The plasmids described here are IncX3 plasmids and they should be referred as that, exclusively. The partial match with ColKP3 found by the plasmid typing scheme is due to the presence of truncated repA gene that matches ColKP3 plasmid rep sequence and is present (as this truncated version) in all X3-OXA-181 plasmids described so far. As per reviewer's knowledge the correct name for the plasmid types described in the current manuscript should be IncX3 instead of IncX3/ColKP3.
2. Line 31: "originated" instead of "originating"?
3. Line 68: "Gram-negative" instead of "gram-negative"?
4. Line 89: remove "the" from before "three CP-E. hormaechei"?
5. Line 113: in general AMR Finder gives better accuracy for AMR calling/numbering.
6. Was PlasmidFinder used for plasmid typing? I cannot see the reference.
7. I would suggest moving Lines 127 to 137 after general plasmid descriptions (lines 138-145).
8. When mentioned in lines 132 to 134 the possible explanation for the ISEcp1-OXA181 genomic context (truncated repA flanking this region), please also include following reference "ISEcp1-Mediated Transposition and Homologous Recombination Can Explain the Context of blaCTX-M-62 Linked to qnrB2" (Zong et al), where it was demonstrated this type of mechanism.
9. Lines 140-145: % nucleotide similarity is not correct/informative, so I suggest remove that. Instead, I would include more detailed information/figure to illustrate where exactly those ISs are located. I downloaded the sequences available (only assembled plasmids are available) from NCBI and by comparison of X3 plasmids with AP018831 (given as reference by the authors), there are two extra ISs in pCUVET22-969.1 (and some extra nucleotides flanking those ISs and interrupting virB1) and rest of plasmid is identical to AP018831. However, fig 2 seems to have lost the ORF upstream of ISKpn74 and does not show virB1 interruption by IS1294. Please, also check properly IS annotation with IS Finder, look for DR/IR as I have done in point 10 below, these sequences could be further analyzed. Figure 2 could be more specific on those changes as the rest of the plasmid is identical to classic X3-OXA-181.
10. When comparing plasmid pCUVET18-1371.4 with AP018831, sequences are also identical except with an extra IS (and some extra bases flanking the IS) and a couple of SNPs that I cannot see reflected in the text (as mentioned in point 9, % similarity is not correct/informative). Also, ISEc1 does not seem truncated, and I could find 14-bp direct repeats flanking this IS (CTTTTTTCAACAGG). If I had the SRA available, I would have double checked for SNPs. I suggest reviewing sequences for the other insertions in pCUVET22-969.1 (point 9. look at IS Finder to properly annotate and identify IR and DR) and re-write paragraph on lines 138-145. Also in text, better to compare pCUVET18-1371.4 to reference AP018831 than pCUVET22-969.1 (lines 143-145).
11. Better discussion of these insertions and some more information of origin/frequency in these genomes would be of great value.
12. Figure 1. Subscript for beta-lactam genes. I would suggest that intrinsic genes and truncated (as ereA) would be removed.
13. Figure 2. Please, review the absence of ORFs from fig as mentioned above (point 9). Also include reference as mentioned above (point 8) and ref 16 (Liu et al) after lines 303 and 305. I would also suggest modifying the sentence (the affirmation is very strong) as this has not been prov

**First Report of *bla*_{OXA-181}-Carrying IncX3/ColKP3 Plasmids in Multidrug-Resistant**
***Enterobacter hormaechei* and *Serratia nevei* Recovered from Canine and Feline**
**Opportunistic Infections**

Chavin Leelapsawas,^a Parinya Sroithongkham,^a Sunchai Payungporn,^b Pattaraporn
Nimsamer,^b Jitrapa Yindee,^a Alexandra Collaud,^c Vincent Perreten,^c Pattrarat
Chanchaithong^{a,d}#

[revised manuscript text omitted]

genes in these bacteria may play important roles for survival within the environment and
animal intestinal tract.

The OXA-181-producing *E. hormaechei* and *S. nevei* strains carried two to eight
plasmids. The *bla*_{OXA-181}-carrying IncX3/ColKP3 plasmid was present in all strains and co-
harbored *qnrS1* and $\Delta ere(A)$ genes (Fig. 2). The resistance genes were situated in a 14-kbp
IS26 pseudo-composite transposon, located downstream of *umuD* (16, 17). The *bla*_{OXA-181}
was preceded by $\Delta ISEcp1$ and followed by $\Delta lysR$ and $\Delta ere(A)$, which were flanked by
fragments of *repA*. A 5-bp AT-rich sequence repeat (ATCTT) was present at the extremities of
$\Delta repA$, suggesting the insertion and subsequent recombination between two copies of *ISEcp1*
of the ColKP3 replicon into this region (16). The *qnrS1* was separately located downstream
of $\Delta repA$ at the right extremity of the transposon. Additionally, these plasmids have
conjugative regions consisting of a relaxase gene (*taxC*), a gene encoding type IV coupling
protein (*virD4*), and a *virB* gene cluster of type IV secretion system (Fig. 2).

Five of the six CP-*Eh* strains carried an identical 51,479-bp *bla*_{OXA-181}-carrying
plasmid, which shared 100% DNA identity to an *E. hormaechei* plasmid (pM206-OXA-181,
GenBank accession number AP018831) isolated from a Burmese patient (18). The
IncX3/ColKP3 plasmid pCUVET22-969.1 of strain CUVET22-969 displayed 94.9%
nucleotide similarity to the others and was larger (54,232 bp) due to the insertion of *ISKpn74*
and *IS1294* into the *virB* region. The 52,830-bp IncX3/ColKP3 plasmids pCUVET18-1371.4
and pCUVET18-1784.4 of CP-*Sn* strains shared 92.6% nucleotide similarity with
pCUVET22-969.1; they had an additional insertion of $\Delta ISEcl1$ upstream of *qnrS1* (
[revised manuscript text omitted]

 identity with the pCUVET18-1784.4 plasmid found in another *S. nevei* strain. The genetic
 environment of *bla*_{OXA-181} flanked by two copies of IS26 composite transposon is
 demonstrated at the bottom. The 5-bp AT-rich duplicate sequences (ATCTT), due to the
 insertion and subsequent recombination between two copies of *ISEcp1*, are present at the
 extremities of *ΔrepA*. Arrows indicate the orientation and length of genes. The color of each
 arrow represents the gene type or function as given in the legend below. The dark grey
 shading indicates 100% sequence similarity. The symbol Δ indicates truncation.

*Ic*, *aac(6)-IIc*, aminoglycoside acetyltransferase; *ant(2'')-Ia*, *ant(3'')-Ia*, aminoglycoside nucleotidyltransferase; *aph(3')-Ia*, *aph(3'')-Ib*, *aph(6)-*
*Id*, aminoglycoside phosphotransferase; *aac(6')-Ib-cr5*, a fluoroquinolone-acetylating aminoglycoside acetyltransferase; *gyrA* (S83I), a point
mutation at position 83 in the quinolone resistance-determining region of *gyrA* which serine (S) was substituted by isoleucine (I) for
fluoroquinolone resistance; *qnrA1*, *qnrB1*, *qnrS1*, DNA gyrase protection genes for quinolone resistance; *fosA*, glutathione S-transferase for
fosfomycin resistance; *arr*, *arr-3*, ADP-ribosyltransferase for rifampicin resistance; *tet(41)*, *tet(A)*, *tet(D)*, tetracycline efflux genes for
tetracycline resistance; *ere(A)*, erythromycin esterase A for macrolide resistance; *lnu(F)*, lincosamide nucleotidyltransferase for lincosamide
resistance; *catA1*, *catA2*, *catB*, *catB3*, *cmlA10*, chloramphenicol efflux genes for chloramphenicol resistance; *floR*, florfenicol/chloramphenicol
efflux gene for florfenicol and chloramphenicol resistance; *sul1*, *sul2*, dihydropteroate synthase for sulfonamide resistance; *dfrA1*, *dfrA14*,
*dfrA19*, dihydrofolate reductase for trimethoprim resistance; *mcr-9.1*, phosphoethanolamine transferase for colistin resistance; *sat2*,
streptothricin acetyltransferase for streptothricin resistance.

**FIG 2** Sequence comparison of the *bla*_{OXA-181}-carrying IncX3/ColKP3 plasmids in
*Enterobacter hormaechei* and *Serratia nevei* isolated from dogs in Thailand with pM206-
OXA181 (GenBank accession number [AP018831](https://www.ncbi.nlm.nih.gov/nuccore/AP018831)) harbored by *E. hormaechei* ST231
isolated from a patient in Myanmar. The same sequence structure of IncX3/ColKP3 plasmid
in *E. hormaechei* ST182 (pCUVET18-121.3) was found in pCUVET19-891.3, pCUVET21-
1190.1, pCUVET21-1726.1, and pCUVET22-793.2, all exhibiting 100% nucleotide identity.
The IncX3/ColKP3 plasmid in *S. nevei* (pCUVET18-1371.4) also showed 100% nucleotide
identity with the pCUVET18-1784.4 plasmid found in another *S. nevei* strain. The genetic
environment of *bla*_{OXA-181} flanked by two copies of IS26 composite transposon is
demonstrated at the bottom. The 5-bp AT-rich duplicate sequences (ATCTT), due to the
insertion and subsequent recombination between two copies of *ISEcp1*, are present at the
extremities of *ΔrepA*. Arrows indicate the orientation and length of genes. The color of each
arrow represents the gene type or function as given in the legend below. The dark grey
shading indicates 100% sequence similarity. The symbol Δ indicates truncation.

Leelapsawas et al. describe the presence of IncX3-*bla*_{OXA-181} in *E. hormaechei* (n=6) and *S. nevei* (n=2) in dogs and cats in Thailand for the first time. Data are generally presented coherently but analysis and discussion would benefit if some modifications were implemented. Please find below some comments for the authors to consider.

1. The continuing reference to IncX3/ColKP3 plasmid in the title and throughout the manuscript should be avoided. The plasmids described here are IncX3 plasmids and they should be referred as that, exclusively. The partial match with ColKP3 found by the plasmid typing scheme is due to the presence of truncated *repA* gene that matches ColKP3 plasmid *rep* sequence and is present (as this truncated version) in all X3-OXA-181 plasmids described so far. As per reviewer's knowledge the correct name for the plasmid types described in the current manuscript should be IncX3 instead of IncX3/ColKP3.
2. Line 31: "originated" instead of "originating"?
3. Line 68: "Gram-negative" instead of "gram-negative"?
4. Line 89: remove "the" from before "three CP-*E. hormaechei*"?
5. Line 113: in general AMR Finder gives better accuracy for AMR calling/numbering.
6. Was PlasmidFinder used for plasmid typing? I cannot see the reference.
7. I would suggest moving Lines 127 to 137 after general plasmid descriptions (lines 138-145).
8. When mentioned in lines 132 to 134 the possible explanation for the *ISEcp1*-OXA181 genomic context (truncated *repA* flanking this region), please also include following reference "*ISEcp1*-Mediated Transposition and Homologous Recombination Can Explain the Context of *bla*_{CTX-M-62} Linked to *qnrB2*" (Zong et al), where it was demonstrated this type of mechanism.
9. Lines 140-145: % nucleotide similarity is not correct/informative, so I suggest remove that. Instead, I would include more detailed information/figure to illustrate where exactly those ISs are located. I downloaded the sequences available (only assembled plasmids are available) from NCBI and by comparison of X3 plasmids with AP018831 (given as reference by the authors), there are two extra ISs in pCUVET22-969.1 (and some extra nucleotides flanking those ISs and interrupting *virB1*) and rest of plasmid is identical to AP018831. However, fig 2 seems to have lost the ORF upstream of *ISKpn74* and does not show *virB1* interruption by *IS1294*. Please, also check properly IS annotation with IS Finder, look for DR/IR as I have done in point 10 below, these sequences could be further analysed. Figure 2 could be more specific on those changes as the rest of the plasmid is identical to classic X3-OXA-181.
10. When comparing plasmid pCUVET18-1371.4 with AP018831, sequences are also identical except with an extra IS (and some extra bases flanking the IS) and a couple of SNPs that I cannot see reflected in the text (as mentioned in point 9, % similarity is not correct/informative). Also, *ISEc1* does not seem truncated, and I could find 14-bp direct repeats flanking this IS (CTTTTTTCAACAGG). If I had the SRA available, I would have double checked for SNPs. I suggest reviewing sequences for the other insertions in pCUVET22-969.1 (point 9. look at IS Finder to properly annotate and identify IR and DR) and re-write paragraph on lines 138-145. Also in text, better to compare pCUVET18-1371.4 to reference AP018831 than pCUVET22-969.1 (lines 143-145).
11. Better discussion of these insertions and some more information of origin/frequency in these genomes would be of great value.
12. Figure 1. Subscript for beta-lactam genes. I would suggest that intrinsic genes and truncated (as *ereA*) would be removed.

13. Figure 2. Please, review the absence of ORFs from fig as mentioned above (point 9). Also include reference as mentioned above (point 8) and ref 16 (Liu et al) after lines 303 and 305. I would also suggest modifying the sentence (the affirmation is very strong) as this has not been proved but suggested by Liu et al.

Manuscript – Spectrum03589-23

The authors are very grateful to the reviewers for their thoughtful consideration and valuable comments, which have significantly contributed to the enhancement of our manuscript. Each comment has been addressed, and the manuscript has been revised in a point-by-point manner. The line numbers used in the response to reviewers correspond consistently with those in the marked-up manuscript file.

Response to reviewer #1 comments

The manuscript reports interesting data on the dissemination of multi-drug resistance plasmids in enteric bacteria recovered from companion animals in Thailand. Considering the limited information regarding plasmid circulation in animals, this work despite its descriptive nature contributes to a better understanding of AMR epidemiology in these reservoirs. The genomic analysis is detailed and well presented. I suggest providing some information on sequencing metrics and a better description in the text of the other MDR plasmids identified in these isolates instead of just listing them in Table S1. Find some other minor suggestions in the annotated pdf.

Point #1 – Line 37: change “minor variant” to variation.

Response #1 – “minor variant” has been changed to “variation”

Point #2 – Line 38-40: Revise the sentence “Additionally, the *bla*_{OXA-181} plasmids of *S. nevei* strains displayed nearly identical to the others, but they had the insertion of Δ *ISEcII* upstream of the *qnrS1* gene.

Response #2 – The sentence has been changed to “Additionally, the *bla*_{OXA-181} plasmids of *S. nevei* strains were nearly identical to the others at the nucleotide level, with *ISEcII* inserted upstream of the *qnrS1* gene.”.

Point #3 – Line 42: Remove “their”

Response #3 – “their” in the sentence is removed.

Point #4 – Line 104: Correct “Whole genome sequence” to “Whole genome sequences”.

Response #4 – The sentence has been corrected to “Whole genome sequences were obtained ...”.

Point #5 – How was the quality of the output? Were these genomes closed or how many contigs after assembly for each? Have these reads/sequences been deposited in a database?

Response #5 – We additionally provide Table S3 in supplemental materials to illustrate the quality of the output, number of closed circular contigs, and nucleotide accession number. Table S3 is additionally referred in Line 109. The complete genome sequences have been deposited in

NCBI database under BioProject accession number PRJNA912905 as indicated in the Data availability section.

Point #6 – Line 114: Add a reference about the ST171 high-risk clone.

Response #6 –Reference 16 (Gomez-Simmonds A et al. 2018. Genomic and geographic context for the evolution of high-risk carbapenem-resistant *Enterobacter cloacae* complex clones ST171 and ST78. mBio. 9:e00542–18.) has been added in the sentence in Line 114.

Point #7 – Line 124: Does this differ from other *Enterobacter* strains/species?

Response #7 – The virulence-associated genes found in CP-*Eh* strains in this study were compared to those in other *Enterobacter* strains or species in a previous study from Bolourchi et al., 2022. Therefore, we explain with the additional sentence **“The CP-*Eh* strains exhibited a higher number of virulence-associated genes, compared to other *Enterobacter* species or strains in a previous study (17).”** in Lines 124 with the reference.

Point #8 – Are these plasmids mobilizable?

Response #8 – We has modified the sentence to **“Additionally, these plasmids were mobilizable, as they had conjugative regions consisting of a relaxase gene (*taxC*), a gene encoding type IV coupling protein (*virD4*), and *virB* gene cluster of type IV secretion system (Fig. 2) (18).”** and add reference 18 that present transferability of the plasmids. The sentences are moved to the sentence to Line 146 according to reviewer#2’s suggestion)

Point #9 – Line 152: Unclear statement

Response #9 – The sentence has been re-written to **“The detection of this resistance plasmid in *Enterobacter* and *Serratia* is relatively rare, as these genera causing opportunistic diseases are less commonly found when compared to *E. coli* and *K. pneumoniae* (2, 17).”** in Lines 152

Point #10 – Line 175-176: I think you need to describe these a little more here - were the AMR genes clustered? associated with transposons? were the clusters all the same in the plasmids from different isolates?

Response #10 – We describe more about additional MDR plasmids carried in CP-*Eh* and CP-*Sn* strains for referring to antimicrobial resistance gene localization in Table S1.

“Two additional MDR plasmids were detected in CP-*Eh* ST182 strain, including IncFIB and IncR plasmids, which contained 13 and four ARGs, respectively (Table S1).” has been added in Lines 154.

“Additionally, two CP-*Sn* strains harbored identical IncA/C2 and IncFII MDR plasmids, each containing eight and six ARGs, respectively (Table S1).” has been added in Lines 162.

Response to reviewer #2 comments

Leelapsawas et al. describe the presence of IncX3-*bla*_{OXA-181} in *E. hormaechei* (n=6) and *S. nevei* (n=2) in dogs and cats in Thailand for the first time. Data are generally presented coherently but analysis and discussion would benefit if some modifications were implemented. Please find below some comments for the authors to consider.

Point #1 – The continuing reference to IncX3/ColKP3 plasmid in the title and throughout the manuscript should be avoided. The plasmids described here are IncX3 plasmids and they should be referred as that, exclusively. The partial match with ColKP3 found by the plasmid typing scheme is due to the presence of truncated repA gene that matches ColKP3 plasmid rep sequence and is present (as this truncated version) in all X3-OXA-181 plasmids described so far. As per reviewer's knowledge the correct name for the plasmid types described in the current manuscript should be IncX3 instead of IncX3/ColKP3.

Response #1 – The term “IncX3/ColKP3” has been revised to “IncX3” in the title and consistently applied throughout the manuscript.

Point #2 – Line 31: "originated" instead of "originating"?

Response #2 – The word “originating” has been substituted with “originated” in Line 32.

Point #3 – Line 68: "Gram-negative" instead of "gram-negative"?

Response #3 – The word “gram-negative” has been substituted with “Gram-negative” in Line 68.

Point #4 – Line 89: remove "the" from before "three CP-*E. hormaechei*"?

Response #4 – “the” is removed from the sentence in Line 89.

Point #5 – Line 113: in general AMR Finder gives better accuracy for AMR calling/numbering.

Response #5 – The “NCBI AMRFinderPlus v3.11.17” has been added in Line 113.

Point #6 – Was PlasmidFinder used for plasmid typing? I cannot see the reference.

Response #6 – In line 113, we add the sentence “Plasmid incompatibility complex (Inc) groups and insertion sequence (IS) elements were identified by PlasmidFinder v2.1 and ISFinder tools.”.

Point #7 – I would suggest moving Lines 129-137 after general plasmid descriptions (line 145).

Response #7 – The sentences describing genetic environment of *bla*_{OXA-181} are moved from Lines 129-137 to Lines 146 after the general plasmid description.

Point #8 – When mentioned in lines 133 to 135 the possible explanation for the ISE*cpI*-OXA181 genomic context (truncated *repA* flanking this region), please also include following reference

"*ISEcpI*-Mediated Transposition and Homologous Recombination Can Explain the Context of *bla*_{CTX-M-62} Linked to *qnrB2*" (Zong et al), where it was demonstrated this type of mechanism. Response #8 – Reference 22 (Zong et al. 2010. *ISEcpI*-mediated transposition and homologous recombination can explain the context of *bla*_{CTX-M-62} linked to *qnrB2*. *Antimicrob Agents Chemother* 54:3039–42.) has been added in Line 146.

Point #9 – Lines 140-145: % nucleotide similarity is not correct/informative, so I suggest remove that. Instead, I would include more detailed information/figure to illustrate where exactly those ISs are located. I downloaded the sequences available (only assembled plasmids are available) from NCBI and by comparison of X3 plasmids with AP018831 (given as reference by the authors), there are two extra ISs in pCUVET22-969.1 (and some extra nucleotides flanking those ISs and interrupting *virB1*) and rest of plasmid is identical to AP018831. However, fig 2 seems to have lost the ORF upstream of *ISKpn74* and does not show *virB1* interruption by *IS1294*. Please, also check properly IS annotation with IS Finder, look for DR/IR as I have done in point 10 below, these sequences could be further analyzed. Figure 2 could be more specific on those changes as the rest of the plasmid is identical to classic X3-OXA-181.

Response #9 – Lines 140-145 have been revised to “**The 54,232-bp IncX3 plasmid pCUVET22-969.1 of strain CUVET22-969 had a larger size than the others due to the interruption of *virB1* by *ISSboI* and the insertion of *ISKpn74* between *virB10* and *virB11*. The *ISKpn74* had two 17-bp inverted repeats, each preceded by a 9-bp direct repeat (DR). The 52,830-bp IncX3 plasmids pCUVET18-1371.4 and pCUVET18-1784.4 of CP-*Sn* strains had a few SNPs compared to pM206-OXA-181 and had an additional insertion of *ISEcII* upstream of *qnrS1*, flanked by two 14-bp DRs (Fig. 2).**”

The annotation of “*IS1294*” has been corrected to “*ISSboI*” in the text, figure 2, and figure 2 legend according to the highest score of BLAST search result from ISFinder tool.

An ORF upstream of *ISKpn74* and DRs/IRs flanking *ISKpn74* are indicated in figure 2.

Point #10 – When comparing plasmid pCUVET18-1371.4 with AP018831, sequences are also identical except with an extra IS (and some extra bases flanking the IS) and a couple of SNPs that I cannot see reflected in the text (as mentioned in point 9, % similarity is not correct/informative). Also, *ISEcII* does not seem truncated, and I could find 14-bp direct repeats flanking this IS (CTTTTTTCAACAGG). If I had the SRA available, I would have double checked for SNPs. I suggest reviewing sequences for the other insertions in pCUVET22-969.1 (point 9. look at IS Finder to properly annotate and identify IR and DR) and re-write paragraph on lines 148-145. Also in text, better to compare pCUVET18-1371.4 to reference AP018831 than pCUVET22-969.1 (lines 143-145).

Response #10 – The “ Δ ISEcII” has been corrected to “ISEcII” in the text and figure 2. The 14-bp DRs of the ISEcII are indicated in figure 2. This paragraph has been re-written (Lines 129-137)

The IncX3 plasmids in figure 2 have been rearranged to compare pCUVET18-1374.4 with the reference pM206-OXA-181 (GenBank accession number AP018831) as described in the text (Lines 140).

Point #11 – Better discussion of these insertions and some more information of origin/frequency in these genomes would be of great value.

Response #11 – The sentence **“Presence of IS originally identified in other Enterobacteriales and their absence in the CP-Eh ST171 genomes suggest that plasmid pCUVET22-969.1 has been transiting in other bacteria prior acquisition by *E. hormaechei*, also highlighting promiscuous potential of IncX3 plasmids for carbapenemase dissemination.”** is added in Line 143.

The sentence **“ISs can function as molecular hotspots, facilitating the acquisition of additional genes and promoting recombination for the evolution of plasmids (19).”** is added in Line 145.

Point #12 – Figure 1. Subscript for beta-lactam genes. I would suggest that intrinsic genes and truncated (as *ereA*) would be removed.

Response #12 – In figure 1, the beta-lactamase genes have been rewritten in subscripts, and intrinsic resistance genes and Δ *ereA* are removed from Figure 1 and Table S1.

Point #13 – Figure 2. Please, review the absence of ORFs from fig as mentioned above (point 9). Also include reference as mentioned above (point 8) and ref 16 (Liu et al) after lines 348 to 350. I would also suggest modifying the sentence (the affirmation is very strong) as this has not been proved but suggested by Liu et al.

Response #13 – Additional ORF, IS elements and repeat sequences have been indicated as described in Response #9 and #10.

The sentence in Line 350 of figure 2 legend has been revised to **“The 5-bp AT-rich duplicate sequences (ATCTT) of the extremities of Δ *repA* suggested insertion and subsequent recombination between two copies of ISEcp1 (20, 22).”**

The references numbers 20 and 22 (Liu et al., 2015 and Zong et al., 2010) are included.

Re: Spectrum03589-23R1 (First Report of *bla*_{OXA-181}-Carrying IncX3 Plasmids in Multidrug-Resistant *Enterobacter hormaechei* and *Serratia nevei* Recovered from Canine and Feline Opportunistic Infections)

Dear Dr. Pattrarat Chanchaithong:

Your revised manuscript has been accepted, and I am forwarding it to the ASM production staff for publication. Your paper will first be checked to make sure all elements meet the technical requirements. ASM staff will contact you if anything needs to be revised before copyediting and production can begin. Otherwise, you will be notified when your proofs are ready to be viewed.

Sincerely,
Ruth Hall
Editor
Microbiology Spectrum